# A Propagation Study of LoRa P2P Links for IoT Applications: The Case of Near-Surface Measurements over Semitropical Rivers

**DOI:** 10.3390/s21206872

**Published:** 2021-10-16

**Authors:** Amado Gutiérrez-Gómez, Víctor Rangel, Robert M. Edwards, John G. Davis, Raúl Aquino, Jesús López-De la Cruz, Oliver Mendoza-Cano, Miguel Lopez-Guerrero, Yu Geng

**Affiliations:** 1School of Engineering, National Autonomous University of Mexico (UNAM), Building Q-Valdes Vallejo, Ciudad Universitaria, Delegación de Coyoacán, CDMX, Mexico City 04510, Mexico; victor@fi-b.unam.mx; 25G Research Centre, Loughborough University, Loughborough LE11 3TU, UK; R.M.Edwards@lboro.ac.uk (R.M.E.); J.G.Davis@lboro.ac.uk (J.G.D.); Y.Geng@lboro.ac.uk (Y.G.); 3School of Telematics, University of Colima, 333 University Avenue, Colima 28045, Mexico; aquinor@ucol.mx; 4Faculty of Civil Engineering, University of Colima, km. 9, Carretera Colima-Coquimatlán, Col. Jardines del Llano, Coquimatlán, Colima 28400, Mexico; jlopez@ucol.mx (J.L.-D.l.C.); oliver@ucol.mx (O.M.-C.); 5Department of Electrical Engineering, Metropolitan Autonomous University, Iztapalapa 09340, Mexico; milo@xanum.uam.mx

**Keywords:** chirp modulation, internet of things, path loss

## Abstract

Internet of Things (IoT) radio networks are becoming popular in several scenarios for short-range applications (e.g., wearables and home security) and medium-range applications (e.g., shipping container tracking and autonomous farming). They have also been proposed for water monitoring in flood warning systems. IoT communications may use long range (LoRa) radios working in the 915 MHz industrial, scientific and medical (ISM) band. In this research, we study the propagation characteristics of LoRa chirp radio signals close to and over water in a tropical meadow region. We use as a case study the Colima River in Mexico. We develop a novel point-to-point IoT measurement sounding system that does not require decoding of LoRa propriety bursts and provides accurate power versus distance profiles along the riparian zone of a steeply dropping mountain river. We used this system to obtain the measurements reported in this work, which are also analyzed and modeled. The results show that the LoRa signal propagation over water exhibits a log-normal distribution. As a result of the chirp signal processing, two new experimental path loss models are presented. The path loss results show a considerable degradation of the received signal power over water within vegetation and less signal degradation at antenna heights closer to the water surface.

## 1. Introduction

For many traditional remote sensing applications, that have previously relied on fixed infrastructure, Internet-of-Things (IoT) alternatives are currently being considered. These applications include emergency communication systems [1], ad hoc public safety networks [2], remote structural health monitoring [3], and long-range (LoRa) solutions for rural applications [4].

Recently, in a previous work, a low-cost system to study river conditions was deployed and tested in Colima, Mexico [5,6,7]. Some sections of such communication system were located close to populated areas, and they were implemented using fixed rugged gauging stations connected to the cloud by means of 3G cellular networks. In contrast, other sections did not have cellular coverage because they were located along the sides of an active volcano or at high altitudes. For these areas, fixed infrastructure was not affordable nor protectable. Since the river lines change periodically, a nomadic solution was needed to be recovered, checked, serviced, and redeployed after each rainy season. Security issues and cost considerations also prevented the system from using a tower and solar panels in some sections of the network. There was, therefore, a need for a low-cost, low-power radio network to connect these parts of the system. In this context, a very attractive alternative for transmitting data related to water sensing in the riparian region of the tropical watercourse is the use of point-to-point (P2P) links with LoRa end-devices (ED) [5].

It has been shown that LoRa technology meets three requirements set by IoT radio applications. Namely, low power consumption, low cost, and the ability to support large-scale deployments [8,9]. In addition, the ad-hoc networking approach [10] has been proposed to establish LoRa P2P links for applications requiring cost-effective deployments composed only of motes and used without gateways, cloud services, and application servers. Under this approach, LoRa end-devices have been used to send a short emergency message comprising 1 Byte of information [1]. Due to these capabilities, LoRa technology is a strong candidate to implement P2P links on IoT networks deployed over irregular terrains with difficult access.

Despite these advantages, the planning of LoRa ED P2P networks presents a significant challenge in areas that are hard-to-reach by humans. The planning of a LoRa ED P2P network in rivers within dense vegetation is not a trivial task, since there is a lack of studies of its radio propagation characteristics in these environments.

The main goal of the work reported herein is to contribute to the deployment of such systems. To this end, we present a propagation study for communications occurring in the 915 MHz ISM band in a river environment within dense vegetation. We configured a P2P communication link composed of a real LoRa ED transmitter and a software-defined radio (SDR) receiver. The LoRa radio is a low-cost programmable device, which works with compressed high-intensity radar pulses (CHIRP) at the physical layer. This sounding system allows us to capture the degradation effects of the chirp spread spectrum (CSS) proprietary modulation from LoRa ED caused by water and vegetation. From the received CHIRP signal power, we compute and report the parameters required by path loss models. One of the relevant conditions of our tests is that the communication devices are placed in such a way that their antennas are placed near the water surface, in order to reduce significantly the effects radiated RF signals to be intercepted or absorbed by leaves, twigs, and long branches, as normally found in tropical areas along rivers.

It is worth mentioning that available propagation models (e.g., Okamura Hata [11], Erceg [12], and Walfisch-Bertoni [13]) place transmission antenna at the base station at very high altitudes above 30 m, for these urban models the higher the transmission antennas the lower the propagation loss, since most objects (such as houses and small buildings) are avoided, as can be seen in Figure 1a. However, for the river environment studies in this work with dense vegetation according to [14] even a single tree (with leaves, twigs, and branches) as shown in Figure 1b, will lead to an excess loss of 20 dB at the canopy layer, and 8 dB excess loss at the trunk layer, thus making it more convenient to place the antennas at the trunk layer.

In contrast, having high heights transmission antennas, as used in urban models, will result in an unaffordable solution for a river or forest environment, since this will increase the expenditure significantly for a low-cost IoT system such as an ad-hoc or mesh sensor network with P2P links [10].

Therefore, for this type of river environment, the optimum system performance will be obtained with antenna transmission and reception altitudes lower than the canopy layer at 0–2 m according to the real scenario. This will reduce signal fading caused by multipath propagation and the link distance will be increased. For example, in wireless sensor networks in agricultural and forestry environments, placing the nodes at trunk level will maximize power with less attenuation [14].

Therefore, current propagation models are not applicable to the environmental setting that we are considering.

The main contributions of this paper can be summarized as follows:◦The development of an algorithm to obtain the radio-frequency (RF) power of the received chirp signals from a LoRa ED.◦Estimation of the empirical distribution of the chirp signal propagation over water within vegetation. We demonstrate that the signal propagation over water exhibits a log normal distribution.◦Two experimental path loss models derived from samples taken in a real environment at Colima, Mexico. These models consider the received chirps of RF power and the shadowing effect.◦Coverage analysis of the two experimental path loss models for two antenna heights over water within vegetation. In particular, antenna heights of 50 cm and 1 m were selected for the experiments, since they presented the lowest attenuation, compared to other heights above the leaves, twigs and long branches as seen in Figure 2 for the real scenario. In addition, it is shown that the coverage decreases significantly due to the increased path loss in our scenario.

The remainder of this paper is organized as follows. Section 2 provides an overview of radio propagation measurements with LoRa technology in water and vegetation environments. Section 3 explains the LoRa technology and the LoRa PHY structure under a theoretical and experimental approach. Section 4 describes the characteristics of the test bed environment and the equipment used in the LoRa ED P2P link. Section 5 describes the signal processing to compute the RF power of the chirp signal, least squares, and link budget analysis. Section 6 analyzes and compares two experimental path loss models and the free-space path loss model. Finally, Section 7 presents the conclusions of the research.

## 2. Radio Propagation Overview for LoRa ED P2P Networks

In this section, we summarize relevant work related to the problem of characterizing the LoRa signal propagation characteristics in the considered scenario. We start in Section 2.1 by reviewing the suitability of the received signal strength indicator (RSSI) as a measure of RF power. In addition, we report in Section 2.2 and Section 2.3 some research works that have studied the coverage and signal propagation characteristics of the LoRa signal over saltwater (i.e., the sea surface) or vegetated areas.

### 2.1. RSSI as a Measure of Radio-Frequency Power

In the reviewed literature, some works provide empirical propagation models for LoRa motes [1,10,15,16,17] under different conditions. These works use the RSSI as a measure of the received signal power to compute the path loss exponent (PLE). The RSSI is determined from the bit error rate, typically in 128 steps of 1 dB each. In spite of its popularity, it has been shown [18] that the use of RSSI as an estimate of received power yields inaccurate PLE values. In addition, this method incurs in loss of information details [10] and anisotropic behavior [19] in propagation studies. However, just a few works have tried to make use of an adjustment factor to convert from RSSI values to RF power to compute the real received power [20,21].

### 2.2. Propagation Studies of LoRa Technology over Water

In the literature, some studies of the LoRa signal propagation over the sea surface have been reported. Reference [17] reports some tests carried over the sea with a LoRa radio operating at 868 MHz. The communication link achieved a coverage range of 30 km over saltwater with a line of sight (LOS) and a receiver antenna height (RAH) of 24 m. In addition, a PLE of 1.76 was computed from RSSI values, and according to [18], it can be considered as an accurate value. In [22], some tests were performed at 433 MHz and 858 MHz with LoRa links over seawater. For both frequencies with LOS, the coverage radius was 22 km. Furthermore, for a frequency of 434 MHz with LOS and costly and bulky antennas, a range of 28 km was achieved. In [23], tests at 434 MHz had coverage of 4 km between a gateway installed on a building terrace at a height of 3 m above the sea level and the other end with a transmitter antenna height (TAH) of 80 cm over the sea level on a moving boat with non-LOS conditions (NLOS). In [24], tests at 868 MHz achieved 12.969 km of coverage between a gateway with RAH of 13.2 m (60% clearance of the first Fresnel Zone) ashore and two antennas with TAHs of 2.1 m and 3.5 m above sea level placed on a moving boat. In [25], tests at 433 MHz had a coverage of 400 m between a base station (with an antenna height of 4 m) without obstacles and a LoRa module on a moving boat (with an antenna height of 1.5 m over the water level). In [26], the authors implemented a water environmental mobile observer (WeMo) to measure physical and chemical parameters on lakes and lagoons. They used a LoRa module for data transmission on four rivers with LOS. However, there is no information related on propagation studies or characterization of the environment.

Table 1 shows a comparison of the articles analyzed by different parameters such as coverage, frequency, path loss at a reference distance (PL-d0), reference distance (d0), PLE, TAH, RAH, environment, and RSSI calibration.

According to the reviewed literature, there are no propagation studies over water by using the LoRa technology at a frequency of 915 MHz. In available works all signal propagation studies were made over the sea (saltwater). Moreover, the RAH was often high and placed ashore, whereas, on the other end, the TAH was often low and mounted on a moving boat. In addition, in the studies mentioned above, the measurements were collected using a moving boat, and therefore, it was not possible to obtain several readings at the same location to compute a local average power.

### 2.3. Propagation Studies of LoRa Technology in Vegetated Areas

The case study considered in this paper is the LoRa signal propagation in a mixed scenario of water within vegetation. Up to the authors’ best knowledge, there are no studies that consider this specific combination. However, some works have considered vegetated environments. These articles are described in this subsection considering path loss analysis, coverage, antenna heights, and carrier frequencies.

In [10], coverage of 716 m and a PLE of 2.03 were obtained for a LoRa P2P link in a forest environment at 868 MHz with an antenna height of 1.5 m. The authors observed that the trees behaved as waveguides. Due to this reason, they obtained a PLE value close to the free-space PLE of 2.0. In [15], a propagation study was made in a rural environment (i.e., comprising vegetation, trees, and mountains) at 868 MHz for a LoRa ED-gateway link. The study was carried out at different antenna heights of the LoRa ED of 20 cm, 1.5 m, and 3 m, whereas the gateway was placed at an altitude of 70 m above the ground level. The results for NLOS showed a PLE value of 3.03, in addition, the path loss value increased as the antenna height of the LoRa ED decreased. In [16], the path loss model at 915 MHz for a riparian forest terrain within a rural area with trees and shrubs was computed. With a configuration of P2P LoRa radios, the results showed coverage radius of 1.6 km and 1.5 km and a PLE of 1.12 (data rate 0) and a PLE of 1.05 (data rate 3), respectively. In [27], a device-to-device communication test carried at 915 MHz with LoRa radios reached a coverage radius of 250 m in a forest. The communication link presented an intermittent connection because LoRa is quite sensitive to obstacles and reflections. In [28], the authors made propagation tests across a forest with a LoRa gateway at a RAH of 1.3 m and LoRa ED at different TAHs of 0.5, 1.3, 2, 2.5, and 3 m with a frequency band of 433 MHz. For heights greater than 1.3 m, in summer and spring, the PLE, together with the antenna height, increased due to the natural growth of tree branches and leaves. In [29], the coverage of 1.75 km was reached by a LoRa ED P2P link in a tropical vegetated environment at 868 MHz. The receiver and the transmitter were placed at a height of 25 m and 2 m, respectively. In [30], two tests of coverage at 915 MHz and 868 MHz were made. The first and second tests achieved a range of 1 km and 232 m for an environment within the forest near a lakeside and a forestry terrain, respectively. Regarding measurements on the lake, the authors concluded that some type of reflection over the lake contributed to the results. In [31], coverage studies were performed with LoRa technology at 868 MHz within a vegetated scenario. The coverage varied from 50 m to 90 m because dense vegetation leads to a substantial variability of link quality.

Table 2 shows a comparison of propagation studies in vegetated environments.

According to the reviewed literature, there are various studies focused on the coverage of LoRa ED-gateway links over the sea (saltwater) or within vegetated areas, specifically in the frequency bands of 433 MHz and 868 MHz. Some papers provide empirical propagation parameters such as PLE and PL-d0, which are obtained from their measurement campaigns. However, they did not consider an RSSI calibration procedure to obtain the RF power of the received signal.

We close this section by stressing the fact that there are no available studies on the propagation characteristics of LoRa ED P2P links for a mixed environment of rivers (freshwater) within vegetation, specifically for the 915 MHz frequency band and antenna heights near water.

## 3. LoRa Technology

This paper considered the LoRa ED radio because it operates in the unlicensed industrial, scientific, and medical (ISM) frequency bands under 1 GHz [32]. In addition, the LoRa ED radio uses a spread spectrum modulation and proprietary CHIRP. The CHIRPs and their modulation are collectively known as the chirp spread spectrum (CSS) [33,34]. Also, the LoRa ED uses variable spreading factors that yield coding gain combined with low data rates, low power, and long communication range in the system. This means that LoRa technology is adaptable to IoT applications only require the transmission of small amounts of information [32]. Due to this, LoRa can provide extensive coverage and optimal energy consumption in a communication link defined in terms of the following configuration parameters: spread factor (SF), bandwidth (BW), and coding rate (CR). Due to all these characteristics, the authors of [35,36] have claimed that LoRa is resilient to multipath effects. For this reason, the LoRa technology has been chosen in this paper to study signal propagation in an environment with water and trees.

### 3.1. LoRa PHY Structure

A CHIRP signal increases or decreases in frequency within an interval delimited by the signal BW. The radio PHY structure of LoRa ED is composed of a group of up-chirps and down-chirps characterizing different fields of the transmitted signal, as shown in Figure 3.

The preamble consists of a sequence of unmodulated up-chirps [37] with a programmable number of symbols from 6 to 65,535. This field is used to detect the LoRa packet [38]. The sync field has two purposes, frame symbol synchronization (FSS) and frequency synchronization (FS). FSS uses two up-chirp symbols [39] to identify the LoRa networks. When the decoded sync word does not match the configuration values, the LoRa device will stop listening to the transmission and it will reject the connection [40,41]. The FS uses two down-chirp symbols for frequency synchronization, followed by a down-chirp symbol of 0.25 to denote a silence to synchronize the receiver device in time [39,42]. Regarding the header, there are two configuration modes, namely explicit and implicit. In the explicit mode, the header field provides information about three parameters: payload length, CR, and the presence of an optional cyclic redundancy check (CRC) for the payload. The header has a total size of 20 bits [43], which is always transmitted with a CR of 4/8 [44]. In the implicit mode, the header is disabled; this mode is used for situations where the configuration parameters are already known. The payload field can contain from 1 Byte to 255 Bytes and an optional CRC at the end of the payload [45]. Based on the information mentioned above, different combinations of the radio PHY structure can be configured by the LoRa ED when transmitting information over the river within vegetation. To obtain a propagation model in harsh conditions, the configuration of the radio PHY structure must provide a low time on air and thus a high transmission data rate that allows gathering more samples at each measurement point [23,46]. For this reason, it is necessary to analyze the time on air of the radio PHY structure of Lora technology for different frame configurations.

### 3.2. Frame Characterization of LoRa Transmissions

In this subsection, the radio PHY structure was analyzed under three different configurations:C1: CRC enabled, header enabled, and programmed preamble of eight symbols.C2: CRC disabled, header disabled, and programmed preamble of eight symbols.C3: CRC disabled, header disabled, and without preamble.

All experiments (i.e., C1, C2, and C3) were configured using SF of 7, BW of 125 kHz, and CR of 4/5. According to the well-known equations from [44], the time on air of the LoRa packet was computed for different values of the payload length considering each and every configuration. These results are shown in Figure 4.

As an example of the results shown in this figure, it can be noticed that both configurations C1 and C2, with the same payload length of 10 Bytes, achieve a time on air o 41.216 ms and 36.096 ms, respectively. As another example, when the payload length is 1 Byte (i.e., the smallest payload), configurations C1, C2, and C3 achieve a time on air of 25.856 ms, 20.736 ms, and 12.544 ms, respectively. Consequently, C1 and C2 have higher time on air than C3.

### 3.3. Experimental Validation of the LoRa PHY Structure

The experimental setup used in our experiments comprises an SX1276 LoRa ED as the transmitter and an X310 universal software radio peripheral (USRP) sniffer (SDR) as the receiver. These devices were used to validate the LoRa PHY structure. For each configuration (i.e., C1, C2, and C3) LoRa packets were captured with GNU Radio running in Ubuntu 16.04. The signal processing in all cases was performed in Matlab.

Figure 5 shows the signal processing results for C1, C2, and C3. For C1, the symbol time and preamble time (eight up-chirp symbols) are 1.024 ms and 8.192 ms, respectively. The FSS time (two up-chirp symbols) and the FS time (2.25 down-chirps) are 2.048 ms and 2.304 ms, respectively. The header, payload length (10 Bytes), and CRC are composed of 28 symbols with a length of time of 28.672 ms. Therefore, C1 has a size of 40.15 symbols with a time on air of 41.216 ms. For C2, the time on air is 30.696 ms due to the effect of disabling both CRC and the header. This leads to the removal of five symbols of the LoRa PHY structure. Finally, C3 has two unmodulated up-chirps, two unmodulated down-chirps with a silence of 0.25 symbol, and eight modulated symbols corresponding to 1 Byte of payload length. Due to this, the C3 has a total size of 12.25 symbols with a time on air of 12.544 ms.

According to the analysis just described, C3 is selected for transmission tests over the river within surrounding vegetation. Configuration C3 provides a low time on air and thus, a high data rate that allows gathering more samples at each measurement point.

## 4. Measurement Campaign

### 4.1. Environmental Characteristics

The measurement campaign was performed at Colima River (latitude 19°20′4.30′’ N, longitude 103°40′23.93″ W) in the State of Colima, Mexico, at the beginning of the spring season. This research aims to obtain an empirical propagation model to implement a P2P network adapted to dense vegetation for river monitoring studies. To analyze the propagation effect in a natural environment, both receiver and transmitter were placed over the river. Moreover, the receiver node was placed at a fixed location while the transmitter node was placed at different distances, from 10 m to 130 m in steps of 10 m. Figure 6a presents an aerial view of the measurement locations over a section of the Colima River. In turn, Figure 6b shows a cross-sectional view of the river.

The river flows through an environment that is composed of rocks and dynamic riparian vegetation. The vegetation in this environment is a dry deciduous forest with a height below 20 m. A high percentage of the tree species (75%) lose their leaves during the autumn due to the irregularity of the precipitation.

The species found in the area are enterolobium cyclocarpum, tabebuia rosea, carapa, guianensis, hura polyandra, bursera odorata, cordia elaeagnoides and arbutus unedo. The vegetation in the riverbank is dense; however, during flood events, an important part is removed. The average diameter of the trees varies between 0.5 m and 1.5 m. The Colima River is a perennial water channel with a base flow of 1.5 m^3^/s in the dry season. The channel base is composed of sand and rocks. The maximum watermarks in the studied section indicate that the top width of the river channel varies from 6.5 m on average for the dry season (November-May) to 10 m on average for the rainy season (June to October).

### 4.2. Device Setup and Configuration Parameters

As previously mentioned, in the field we used the SX1276 LoRa ED and the X310 SDR as the transmitter and the receiver, respectively. The LoRa radio was configured with an SF of 7, BW of 125 kHz, and CR of 4/5. The header, preamble, and CRC were disabled. In addition, 1 Byte was used for the payload, which is the minimum length that can be configured to carry information in the LoRa PHY structure. This is enough to carry an emergency warning message (e.g., a flood notification or a help request) [1]. An omnidirectional transmitter antenna was used, specifically a model Pctel Mfd 234 (902–928 MHz) of 3 dBi. At the receiver side, an X310 SDR and the GNU Radio (running on Ubuntu 16.04) software were used to capture and record the signal, respectively. An omnidirectional receiver antenna was used, namely, ANT-126-002 proxicast (700–2700 MHz) of 10 dBi. In addition, a bandpass filter model ZX75BP-915-S+ was placed at the SDR input. Also, a GPSD0 (OCXO) and a 5-volt active GPS antenna were used to synchronize the X310 USRP. The measurements were performed for two different antenna heights (ha): 50 cm (ha = 50 cm) and 1 m (ha = 1 m). Both antennas (i.e., the one at the transmitter and the one at the receiver) were placed at the same height for each scenario. The LoRa ED operated in the 915 MHz band (unlicensed band in Mexico) with a transmit power of 20 dBm. The X310 USRP was configured as a receiver using GNU Radio companion, as shown in Figure 7.

Figure 8 illustrates the environment where the study was carried out. This terrain corresponds to a section of a river in Colima, Mexico. As it can be seen, the river flows through a rocky terrain and it is surrounded by dense vegetation.

## 5. Post-Processing

The SDR captured the transmitted signal of the LoRa ED over the river at a sampling rate of $5\times 10^{6}$ samples per second (SPS). The received signal contains several chirp sequences composed of an in-phase (I) component and a quadrature (Q) component with a guard interval between consecutively received chirp sequences (i.e., frames). This is intended to facilitate the detection of the start of each received frame and to compute the signal-to-noise ratio (SNR) utilizing the floor noise Each frame has 12.25 symbols and 62,720 samples, where each symbol has a duration of 1.024 ms.

### 5.1. Computing the Received RF Power

To obtain the RF power, the peak voltage is computed as follows:(1)$$V_{peak}=\sqrt{I^{2}+Q^{2}},$$
where *I* is the in-phase component and *Q* is the quadrature component. Then, the root mean square (*RMS*) voltage can be obtained as follows:(2)$$V_{RMS}=\sqrt{\frac{I^{2}+Q^{2}}{2}},$$
this voltage across an impedance (*Z*) causes an RF power value given by:(3)$$P=\frac{V_{RMS}^{2}}{Z}=\frac{I^{2}+Q^{2}}{2Z}.\;$$

To obtain a mean chirp frame and thus compute the RF power at each distance, the signals were processed using a Python implementation of Algorithm 1 shown below:
**Algorithm 1.** Signal processing for obtaining the mean signal and RF power. **Input:** Spread factor *SF*; bandwidth *BW*; sampling rate *fs*; the number of symbols in each frame *Nsym*; receiver impedance *Z*; number of frames *FrameNumbers*.**Output:** histograms of the average chirp’s signal as a power signal.Initialization: *SF* = 7, *BW* = 125$\times 10^{3}$ [Hz], *fs* = 5 $\times 10^{6}$ [SPS], *Nsym* = 12.25 (C3), *FrameNumbers* = 100 and *Z* = 50 [Ω].(1)Compute the symbol duration (*T_sym_*), time on air (*T_A_*), and the number of samples (*N_S_*) as follows:(4)$$T_{sym}=\frac{2^{SF}}{BW},$$ where *SF* is the spread factor and *BW* is the bandwidth in Hertz. (5)$$T_{A}=N_{sym}\times T_{sym},\;\;$$ where *N_sym_* is the number of symbols in each frame and *T_sym_* is the symbol duration in seconds. (6)$$N_{S}=f_{s}\times T_{A}.$$ where *f_s_* is the sample rate in samples per second and *T_A_* is the time on air in seconds.(2)Convert the captured *I*, *Q* signal from binary to complex:$$I,Q\leftarrow binary\left(data.dat\right).$$
(3)Detect the start of each frame. Convolution is applied to the *I*, *Q* signal using the number of samples from (6) as follows: WindowSize← 1Ns×np.ones1,NS,   
$$signal=\left|I,Q\right|,\;$$
$$\left(signal*WindowSize\right)\left[n\right]\leftarrow \overset{N_{S}-1}{\underset{k=0}{\sum }}signal\left[k\right]WindowSize\left[n-k\right].$$(4)Search the local maximum (peaks) to discard the frames with a considerable attenuation: $$\left[PeaksVector\right]\leftarrow find\_peaks\left(signal*WindowSize\left[n\right]\right).$$(5)Select an empirical threshold (on the peaks) to identify the start of each frame: $$\left[indexes\right]\leftarrow find\left(PeaksVector>threshold*\max\left(PeaksVector\right)\right).$$The threshold parameter for each distance and height was obtained empirically so that the noise did not significantly affect the analyzed signal. These values are shown in Table 3.(6)At the beginning of each frame, determine its corresponding left and right indexes. The aim of this procedure is to create a vector of frames and to avoid the guard interval between each pair of frames:$$LeftIndex\leftarrow PeaksVector\left[indexes\right]-N_{s},$$
$$RightIndex\;\leftarrow LeftIndex+N_{s}$$.(7)Obtain a mean frame from the vector of 100 frames:
     **for**
i=1→FrameNumbers          Frame ←I,QLeftIndexi:RightIndexiFrameVector←FrameVector Frame
     **end**     *FrameVectorMatrixN_S_*[*FrameNumbers*]←*FrameVector*.
$MeanFrame\left[N_{s}\right]\left[1\right]\leftarrow MeanValueForColumns\left(FrameVectorMatrix\right).$
(8)Determine the signal power histograms of the mean frame: $$PowerHistdBm=hist\left[10\times log_{10}\left(\frac{{\left(real\left(MeanFrame\left[n\right]\right)\right)}^{2}+{\left(imag\left(MeanFrame\left[n\right]\right)\right)}^{2}}{2Z}\right)+30\right].$$(9)Compute the RF power of the mean frame: $$RFPower=mean\left(PowerHistdBm\right).$$


**Table 3 sensors-21-06872-t003:** Thresholds used in step 5 of Algorithm 1.

d (m)	20	30	40	50	60	70	80	90	100	110	120	130
*ha* = 1 m: Threshold	0.87	0.86	0.50	0.66	0.53	0.62	0.58	0.33	0.85	0.76	0.81	0.64
*ha* = 50 cm: Threshold	0.85	0.75	0.88	0.90	0.39	0.65	0.71	0.67	0.85	0.80	0.85	0.76

### 5.2. Experimental Path Loss Model

Algorithm 1 provides the RF power values needed to obtain the experimental path loss models. They include all factors affecting the radio signal in a real scenario. Let us recall that this paper aims to get a simple model for a P2P link of the LoRa radio in a river within a vegetated area with near-water antenna heights without considering other aspects such as tree types and sizes. The path loss of the LoRa signal is experimentally computed as follows:(7)$$\mathrm{PL}={\;P}_{\mathrm{tx}}+G_{\mathrm{tx}}+G_{\mathrm{rx}}-P_{\mathrm{rx}}.$$
where PL is the path loss in dB, P_tx_ is the transmission power in dBm, G_tx_ is the gain of the transmitter antenna in dBi, G_rx_ is the gain of the receiver antenna in dBi, and P_rx_ is the received RF power in dBm. The path loss models are computed through the following parameters: P_tx_ of 20 dBm, G_tx_ of 3 dBi, G_rx_ of 10 dBi, and the RF power in dBm is taken from the mean value for each distance. These values are obtained from the power histograms.

### 5.3. Linear Regression Using the Gradient Descent Technique

A linear regression (LR) analysis is applied to the experimental path loss values to get the path loss exponent (PLE) by means of the gradient descent (GD) technique, commonly used in machine learning and neural networks [47,48].

LR is used to fit the collected data to a straight line of the form y=mx+b. Based on this, LR searches for the coefficients [*m*,*b*] that minimize the empirical loss (Loss). The Loss parameter can be defined as follows:(8)$$Loss\left(m,b\right)=\frac{1}{N}\overset{N}{\underset{i=1}{\sum }}{\left(y_{i}-\left(mx_{i}+b\right)\right)}^{2},$$
where *N* is the number of points in the dataset, and $y_{i}$ is an element of the vector containing the measurement dataset. The GD technique minimizes the empirical loss by gradient descent [49] as follows:(9)$$m\leftarrow m-\alpha \frac{\partial }{\partial m}Loss\left(m,b\right);\;b\leftarrow b-\alpha \frac{\partial }{\partial b}Loss\left(m,b\right),$$
consequently:(10)$$m\leftarrow m+\alpha \left[\frac{2}{N}\overset{N}{\underset{i=1}{\sum }}x_{i}\left(y_{i}-\left(mx_{i}+b\right)\right)\right],\;b\leftarrow b+\alpha \left[\frac{2}{N}\overset{N}{\underset{i=1}{\sum }}\left(y_{i}-\left(mx_{i}+b\right)\right)\right].$$
where α is the step size usually known as the learning rate, and it can take a value between 0–1 [50]. A high α value has the risk of missing the lowest point of the slope. The GD algorithm performs two main steps: gradient computation and update of coefficients (*m*, *b*), which starts with the initialization of α, initial vector of [*m*,*b*] and the total number of iterations, then the algorithm stops when the termination criterion is meet, or just the overfitting starts [51,52]. In this paper, the GD algorithm stops when the shadowing achieves a mean value of 0 and random behavior.

### 5.4. Shadowing

The shadowing effect is the signal strength degradation due to obstacles between the transmitter and the receiver. To examine the shadowing between the LoRa ED and the SDR at different distances over the river with the presence of vegetation, the shadowing is computed experimentally [53] as follows:(11)$$X_{i}=\mathrm{PL}\left(d_{i}\right)-{\mathrm{PL}}_{i}$$
where the PL(d_i_) is the path loss according to the one-slope model at a distance d_i_ in dB and the PL_i_ is the i-th measured path loss sample taken at a distance d_i_.

### 5.5. Link Budget Analysis

The demodulator in the LoRa receiver requires a minimum level of SNR to work properly. For this reason, LoRa manufacturers defined a demodulation threshold of −7.5 dB for an SF of 7 [44]. Therefore, the ratio of the received power signal to the total noise power must be greater than the threshold value to ensure a correct demodulation. Consequently, the SNR value in dB [54] is denoted by:(12)$$\mathrm{SNR}=P_{\mathrm{rx}}-N,$$
where P_rx_ is the received power in dBm and N is the total noise power in dBm.

The SNR is a critical parameter that allows obtaining the sensitivity and link budget of the LoRa ED operating as a receiver. The sensitivity (S) in dBm can be expressed as:(13)$$S=-174+10\log_{10}\left(\mathrm{BW}\right)+\mathrm{NF}+\mathrm{SNR},$$
where the first term is the thermal noise in 1 Hz of bandwidth, BW is the receiver bandwidth in Hz, NF is the fixed noise figure of the receiver in dB, and SNR is the signal to noise ratio in dB. The link budget (LB) in dB is denoted by:(14)$$\mathrm{LB}=P_{\mathrm{tx}}-S,$$
where P_tx_ is the transmit power in dBm and S is the receiver sensitivity in dBm.

It is worth mentioning that the minimum theoretical sensitivity is obtained by substituting the following values in (13): BW of 125 kHz, NF of 6 dB, and SNR of −7.5 dB. Consequently, the minimum receiver sensitivity of the LoRa ED is −124.5 dBm.

Another critical parameter is related to the link budget, which is computed from (14). The theoretical link budget is 144.5 dB with a transmit power of 20 dBm and a sensitivity of −124.5 dBm. The link budget value provides key information about the range where the network can operate correctly. With this information, the network designer can make decisions to improve costs and optimize the network.

### 5.6. Theoretical Models of Path Loss and Foliage-Loss

The total path loss (TPL) for a forest terrain must include the excess loss due to vegetation or foliage [55,56]. For this reason, several empirical excess loss models are evaluated in this section. In the literature, there are several models evaluating the excess attenuation because of the presence of foliage between the transmitter and the receiver. The modified exponential decay (MED) models in dB are defined by the following expression [57]:(15)$$L_{\mathrm{MED}}=A\times f^{B}\times d^{c},$$
where f is the operating frequency in Hertz, d is the depth of the foliage in meters, A, B, and C are empirical fitting constants that depend on the type of foliage. The Weissberger model is applied to environments where the ray path is blocked by means of a dense, dry, in-leaf trees [58]. The ITU-R model estimates the signal loss in an environment with vegetation [59]. Both COST 235 and FITU-R provide models of signal loss for scenarios with trees: in-leaf trees and out-of-leaf trees [60,61]. The parameters of each model are presented in Table 4.

The international telecommunication union (ITU) defined a semi-empirical model of maximum attenuation (MA) under the recommendation P.833-9 [62]. This model estimates the attenuation due to vegetation in dB for operation within the interval from 30 MHz to 100 GHz and can be expressed as follows:(16)$$L_{\mathrm{MA}}=A_{m}\left[1-\exp\left(\frac{-d\gamma }{A_{m}}\right)\right],$$
where A_m_ is the maximum attenuation in dB, γ is the specific attenuation for a very short vegetate path in dBm and d is the path length within the woodland. The maximum attenuation (A_m_) in dB is given by:(17)$$A_{m=}A_{1}\times f^{\alpha_{1}},$$
where f is the frequency in MHz and A_1_ and α_1_ are empirical values. Table 5 shows the empirical values used in (16) and (17). These values correspond to terrain with mixed forests.

In order to obtain the TPL in vegetation and water, both the free-space (FSPL) model and the two-ray model are used for modeling the free space. Thus, the FSPL in dB can be expressed as:(18)$$L_{\mathrm{FSPL}}=-10\log\left(\frac{G_{t}G_{r}\lambda^{2}}{{\left(4\pi \right)}^{2}d^{2}}\right),$$
where λ is the wavelength in meters and d is the distance between the transmitter and the receiver in meters. The two-ray model [63] in dB is given by:(19)$$L_{2\mathrm{Ray}}=\left\{\begin{array}{c} L_{\mathrm{FSPL}},\;d<d_{c}\\ \;\;\;L_{\mathrm{PE}},\;d\geq d_{c} \end{array}\right.,$$
where the d_c_ is the crossing distance in meters and the L_PE_ is associated with the plane earth propagation model in dB, which is given by:(20)$$L_{\mathrm{PE}}=20\log\left(\frac{d^{2}}{h_{\mathrm{tx}}h_{\mathrm{rx}}}\right),$$
where the h_tx_ is the transmitter antenna height in meters and h_rx_ is the receiver antenna height in meters. The crossing distance in dB is expressed as:(21)$$d_{c}=\frac{4{\pi h}_{\mathrm{tx}}h_{\mathrm{rx}}}{\lambda },$$

The two-ray model is composed of FSPL until the crossing distance (d_c_) with a rate proportional to d^−2^, after that the path loss rate increases at rate of d^−4^ due to the plane earth propagation model [56]. This behavior indicates a destructive interference because of signal reflected by the ground [64].

## 6. Measurement Results and Analysis

For illustration purposes, Figure 9 shows the results coming from the processing of the received signal using Algorithm 1, for ha of 1 m and 20 m of separation between the end points of the link. Figure 9a shows the results of the convolution stage. The red cross-shaped markers (above 0.2) indicate the start of each frame; this allows extracting the frames from the entire dataset without the guard intervals in between. Figure 9b presents the mean frame of 100 frames extracted from the dataset. However, the mean frame exhibits a noise component due to the near-water signal propagation. Figure 9c shows the spectrogram of the received mean frame to validate it with the corresponding transmitted frame (C3).

### 6.1. Received RF Power

At each measurement point, the power was averaged by using 100 LoRa frames. Figure 10 and Figure 11 show the histograms of signal power for each measurement point for *ha* values of 1 m and 50 cm, respectively. The histograms reveal a symmetrical shape around their mean value.

Note that the horizontal axes are given in logarithmic units (dBm). For this reason, we propose to use a log-normal distribution to model the received signal power. Thus, the chirp signal power was fitted to a log-normal model to capture its behavior using its mean value and variance at each measurement location. The mean value of each histogram is taken as the RF power at each distance. These values are substituted in the received power parameter in (7) to get the experimental path loss models. In Figure 10 and Figure 11, it can be seen that the chirp signal fluctuations fit the log-normal distribution in amplitude. This means that the samples with a considerable degradation have been successfully discarded by the processing of the signal indicated by Algorithm 1. However, when the transmitter LoRa ED moves away from receiver SDR, the mean value of the log-normal model decreases significantly.

On the other hand, the guard interval was processed to compute the background noise power derived from the analyzed environment and to get the SNR at each distance. Table 6 shows the SNR and background noise power (*P_N_*) values at each distance for both *ha* values of 1 m and 50 cm.

Table 6 shows that the SNR decreased 20 dB and 27 dB at ha of 1 m and ha of 50 cm, respectively. The minimum SNR values are 19 dB and 16 dB for ha of 1 m and ha 50 cm, respectively. According to [65], at SNR values >−2 dB, the probability of false-alarm becomes zero while the probability of detection is 1. According to our results, the SNR values are greater than −2dB. This means that the methodology used to compute the received power is correct.

### 6.2. Path Loss

The path loss models are computed by using the mean values coming from the histograms shown in Figure 10 and Figure 11. Now, the path loss models on a LoRa ED P2P link for two antenna heights of 1 m and 50 cm are analyzed to quantify the path loss in the river within vegetation. Figure 12 shows the experimental path loss values indicated by the markers (i.e., the times and plus signs). The straight lines represent the LR model with a GD fit corresponding to the two antenna heights. As expected, it is shown that the experimental path loss rate increases as the log distance increases. The results show that from 20 m to 130 m, the path loss increases in 24 dB and 23 dB for ha values of 1 m and 50 m, respectively. These results show that the slope of the fitted line for ha of 1 m is steeper than the case where ha is 50 cm. This means that there is more degradation of the signal at higher antenna heights above the water. In addition, it can be seen from Figure 12 that the path loss for two antenna heights is higher than the free space path loss in all analyzed distances. This behavior can be attributed to various factors that are present in the environment, such as water, tree branches, and rocks. The signal propagation between the transmitter LoRa ED and receiver SDR is less affected for ha of 50 cm than ha of 1 m. This can be attributed to two aspects: (1) the line-of-sight (LOS) condition is more likely to occur near the water surface because the tree branches and leaves do not hinder the communication link, and (2) the signal reflectivity caused by water.

### 6.3. Shadowing

Figure 13 shows shadowing versus distance. These results show that for both antenna heights, ha of 1 m and 50 cm, the shadowing presents a random behavior. Consequently, the shadow process can be parameterized as a set of zero-mean Gaussian random variables with different standard deviation (σ) [66] values as follows:(22)$$X_{h1m}\sim \left(0,6^{2}\right),\;20\;m<d<130\;m,$$
and
(23)$$X_{h50\mathrm{cm}}\sim \left(0,4^{2}\right),\;20\;m<d<130\;m.$$

From (22) and (23), it can be seen that, when the antenna is closer to the water, the standard deviation is smaller. This behavior can be attributed to the presence of more objects in the path for ha of 1 m than ha of 50 cm over water. From Figure 13, the shadowing peaks indicate that there are more tree branches and leaves at an ha of 1 m over the water.

Table 7 shows the parameters of the experimental path loss models obtained through the GD technique. The stopping criterion for the GD algorithm was based on the shadowing parameters. That is, a good LS adjustment is achieved when the shadowing signal is random with a mean value of 0 and R^2^ approaches the value of 1. Based on this approach, the R^2^ reaches a value of 1, while the shadowing mean value has a value of 0.

The values from Table 7 can be mapped to a log-distance polynomial model of first-order [67] expressed as follows:(24)$$\mathrm{PL}=\left\{\begin{array}{ll} 17.6+10\times 2.8\times log_{10}\left(\frac{d}{1}\right)+X_{h50\mathrm{cm}}, & ha=50\;\mathrm{cm}\\ 17.8+10\times 2.9\times log_{10}\left(\frac{d}{1}\right)+X_{h1m}, & ha=1\;m \end{array}\right..$$

### 6.4. Link Budget

With the information from Table 7 and Table 8, and Equations (12)–(14), the LB was computed through the parameters from Table 8. The noise over the river within vegetation was obtained using the mean noise of all distances from Table 6.

It is important to highlight that the theoretical link budget has a value of 144.5 dB with a transmission power of 20 dBm and a sensitivity of 124.5 dBm. The free space path loss model is taken as a reference point in the link budget analysis.

Assuming that the experimental path loss model works from 1 m to 33.7 km (see Figure 14), it can be shown that the free-space path loss model reaches a coverage of 33.7 km, while the two experimental path loss models for both ha of 1 m and ha of 50 cm with a noise of −67 dBm achieve a range of 1.3 km (σ of 6 dB) and 1.6 km (σ of 4 dB), respectively. Based on these results, the coverage decreases significantly in 96% and 95% for ha values of 1 m and 50 cm, respectively. This means that there is a degradation of the chirp signal in near-water conditions within vegetation. It is important to consider this behavior of the LoRa ED radio in the design and planning of the network under environments composed of water, tree foliage, and rocks. In addition, the model for ha of 50 cm over the water, the coverage increases up to 19% compared to the model for ha of 1 m over water. This can be attributed to water reflectivity and a more obstacle-free communication link between the transmitter and receiver.

### 6.5. Comparison between Actual Measurements and the Theoretical Path Loss Models

The TPL is composed of the sum of FSPL and excess loss. Under this approach, theoretical TPL is evaluated for two cases. In the first case, TPL is computed by adding the FSPL (from (18)) and the excess loss caused by the presence of vegetation. In the second case, TPL is evaluated by adding the two-ray model (from (19)) and the excess loss due to the vegetation. Both cases were compared to the experimental path loss models from (24).

The mean absolute percentage error (MAPE) was used to measure the difference between the experimental and theoretical models. The MAPE is a statistic measuring the accuracy of the propagation models [55,67] and can be expressed as:(25)$$\mathrm{MAPE}=\frac{1}{N}\overset{N}{\underset{i=1}{\sum }}\left|\frac{\mathrm{EM}-\mathrm{TM}}{\mathrm{EM}}\right|\times 100\%.$$
where EM are empirical model values, TM are theoretical model values and N is the number of samples. It can be assumed that the proposed models and theoretical ones are identical, if the MAPE statistic is less than 10% [68].

Table 9 and Table 10 present the comparison between experimental models and theoretical ones in terms of the MAPE statistic. From Table 9, the results show that for both ha of 1 m and ha of 50 cm, the theoretical models (first case) are not within 10% similarity. Other models such as COST 235 and Weissberger were not taken into account in the comparison because the MAPE statistic turned out to be high. The same behavior is to present in the second case (from Table 10) where the models of two rays plus excess loss do not meet the similarity value of 10%.

On the other hand, other effects such as severe rain were also studied in [69] in the band of 3.5 GHz, where a maximum loss between no rain and severe rain could be up to 5 dB, due to a higher multipath fading caused by wind and rain.

## 7. Conclusions

A testbed campaign to obtain the PL of a LoRa ED P2P link with near-water heights in an environment of a river within vegetation has been carried out. The chirp signals frame from LoRa ED were captured through an SDR to avoid missing information, inaccurate PLE values, and anisotropic behavior.

The time on air of the frame was reduced with the minimum number of symbols to generate frames with a higher data rate. Under this context, the LoRa ED sends frames with the header, CRC disabled and a payload length of 1 Byte. In addition, the received signal was captured with an USRP. The signal processing in Python of the received pure-chirps frames provides the real RF power of the signals. The path loss model was computed employing the real power of the received signals to avoid loss of information due to the demodulation threshold on the receiver. It has been observed that the experimental path loss models for both antenna heights of 1 m and 50 cm compared to the free-space path loss model present an increase of path loss in 20 dB and 18 dB, respectively.

The results of the received sequences over water show that its distribution is log-normal. In addition, the two new experimental path loss models for an environment of a river within vegetation show that the values of PLE and *σ* are smaller at antenna heights closer to water. Using the two new experimental path loss models, the link budget studies at 915 MHz carried out in this paper show that the maximum coverage could be up to 1.3 km and 1.6 km for *ha* values of 1 m and 50 cm, respectively.

This paper proposes a new methodology to obtain propagation models by using the power of pure chirp signals from a LoRa ED radio. This methodology allows avoiding missing information and inaccurate PLE values. Also, the path loss parameters obtained in this paper include the effects of the water and vegetation, therefore, they are valid for scenarios similar to the one analyzed in this work.

The proposed models are based on the real-world chirp signal processing of the LoRa ED under a P2P link. This type of terrain is typical of rivers of North America. The path loss parameters obtained in this paper are intended to be used to design LoRa ED P2P networks by using simulators such as OMNeT++, Riverbed Modeler, or NS3 to study energy consumption, packet data rate, and so forth. Hence future work is to use the experimental path loss model in a discrete event simulator to obtain several performance metrics. These metrics will be fundamental to provide a reliable network design for rivers within vegetation in remote areas of Colima, Mexico.

## Figures and Tables

**Figure 1 sensors-21-06872-f001:**
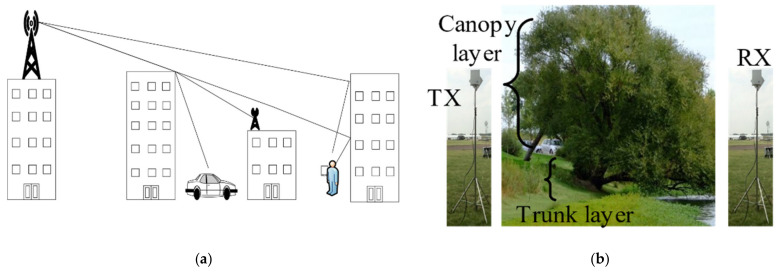
(**a**) Scenario for urban path loss models. (**b**) River environment with large trees.

**Figure 2 sensors-21-06872-f002:**
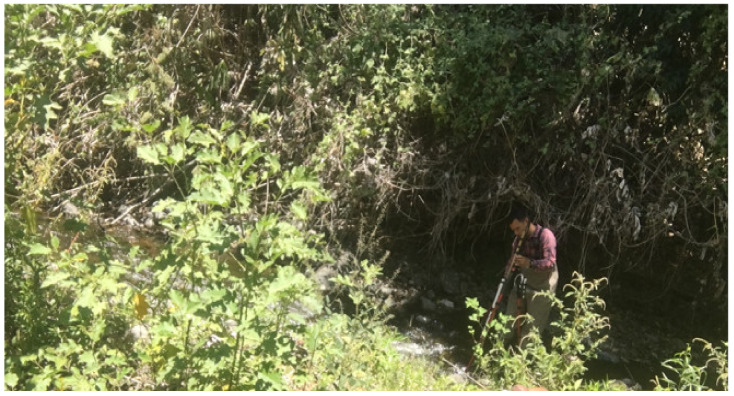
Real scenario for the propagation study. Leaves, twigs, and long branches were constant above 1.5 m over the river.

**Figure 3 sensors-21-06872-f003:**
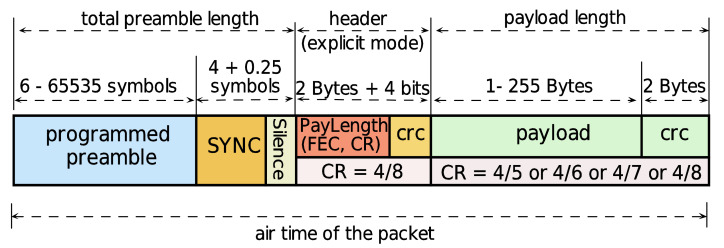
The radio PHY structure is composed of several fields that can be transmitted by the LoRa ED radio.

**Figure 4 sensors-21-06872-f004:**
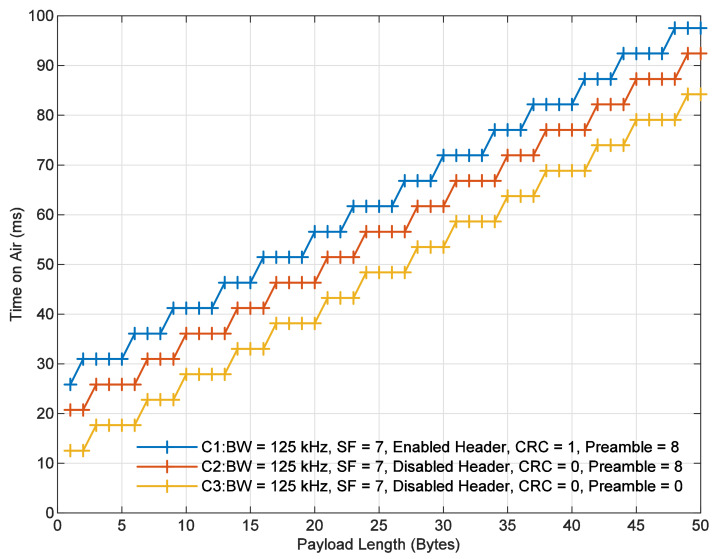
Time on air of the LoRa PHY structure for the analyzed configurations, C1, C2, and C3.

**Figure 5 sensors-21-06872-f005:**
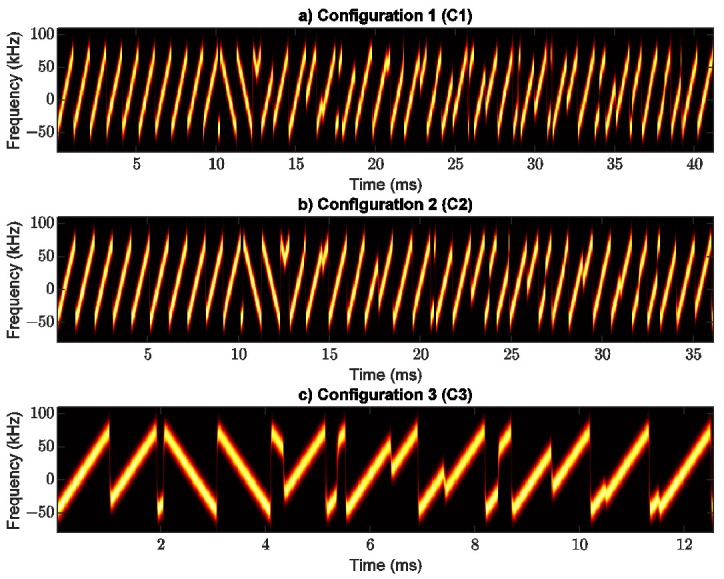
LoRa PHY structure: (**a**) with CRC enabled, header enabled, and programmed preamble of eight symbols, (**b**) with CRC disabled, header disabled and preamble of eight symbols, and (**c**) with CRC disabled, header disabled and without preamble.

**Figure 6 sensors-21-06872-f006:**
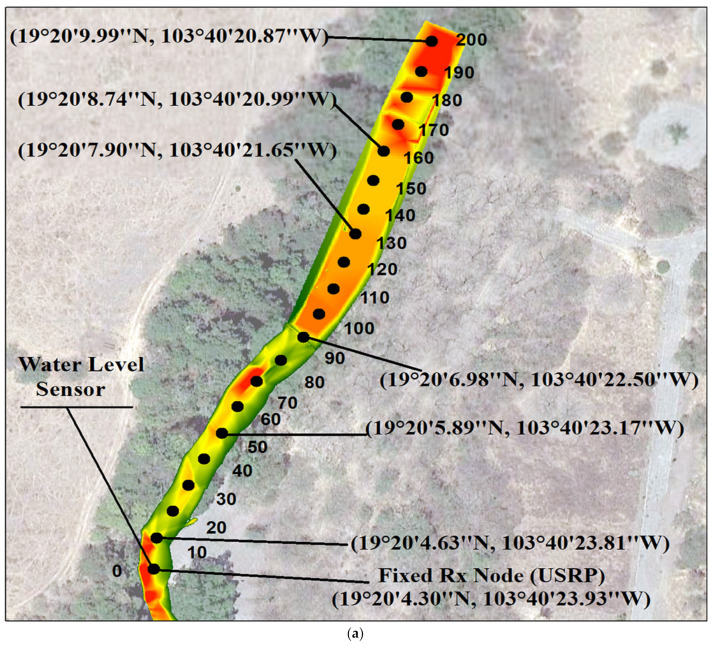
(**a**) Route of the measurement campaign. The receiver node (X310 USRP) is placed at a latitude of 19°20′4.30″ N and longitude of 103°40′23.93″ W whereas the transmitter LoRa ED is repositioned in steps of 10 m (from 10 m to 130 m). (**b**) Elevation along the river route with respect to the sea level, starting at point 0, 835 m above the sea level, and the last point considered in the propagation model was at 0 + 130 m, 838 m above the sea level. Where the LOB and the ROB labels are the Left and Right Over Bank, respectively, and the WS PF1 blue line is the surface of the river at the time to carry out the measurement campaign.

**Figure 7 sensors-21-06872-f007:**
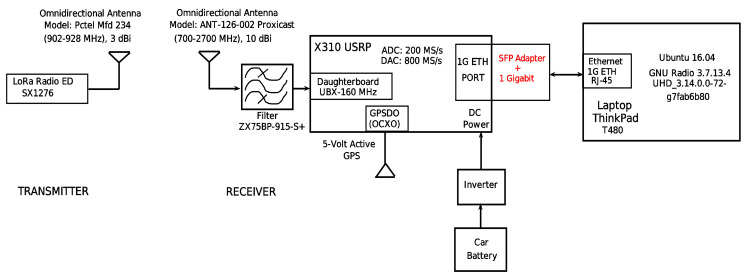
The figure shows the system used for the sounding of the wireless channel. It is composed of a transmitter and a receiver. The LoRa radio was configured as a transmitter whereas in the receiver side the following devices were used: (1) X310 USRP radio, (2) Passband filter to avoid interferences, (3) Laptop to record the received information, and (4) Car battery and inverter to provide electrical current to the receiver radio.

**Figure 8 sensors-21-06872-f008:**
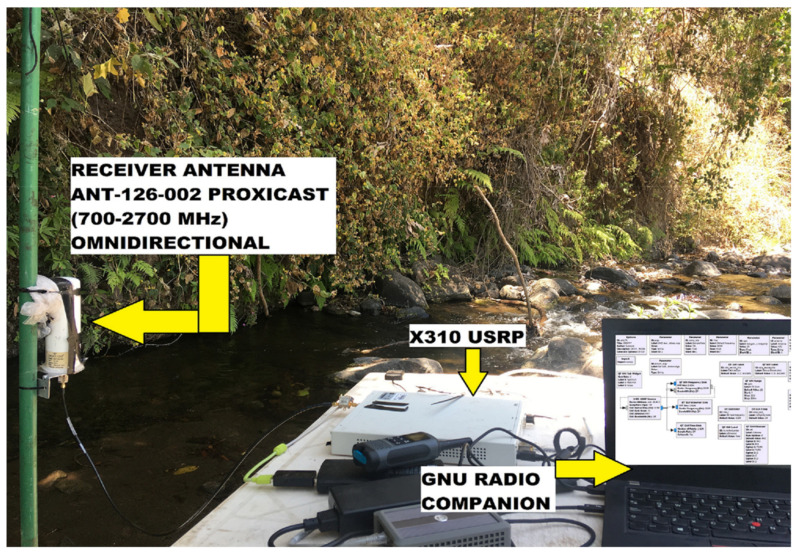
This image shows the devices used in the receiver node: the X310 radio, passband filter, a laptop with GNU Radio, and its corresponding antenna. The surrounding environment is composed of rocks, freshwater, and trees.

**Figure 9 sensors-21-06872-f009:**
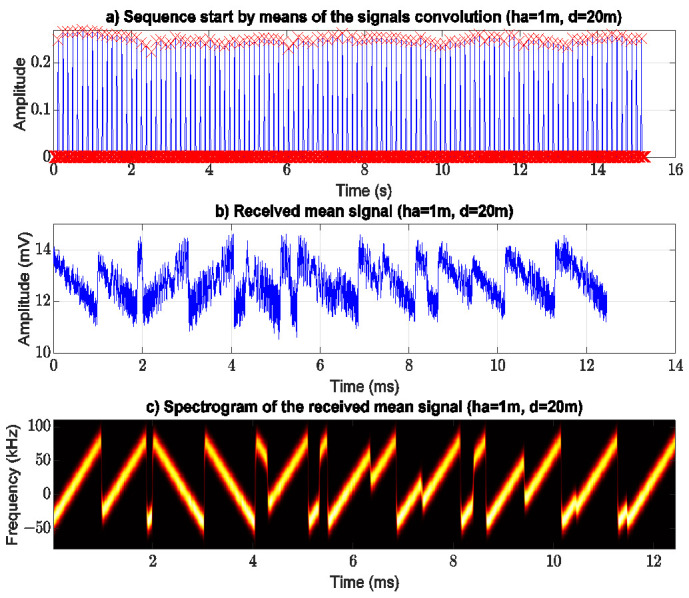
Signal processing to obtain the mean frame of 100 chirp’s frame, (**a**) the red markers show the start of each frame (algorithm 1 point 6), (**b**) mean frame (algorithm 1 point 7), and (**c**) mean frame spectrogram.

**Figure 10 sensors-21-06872-f010:**
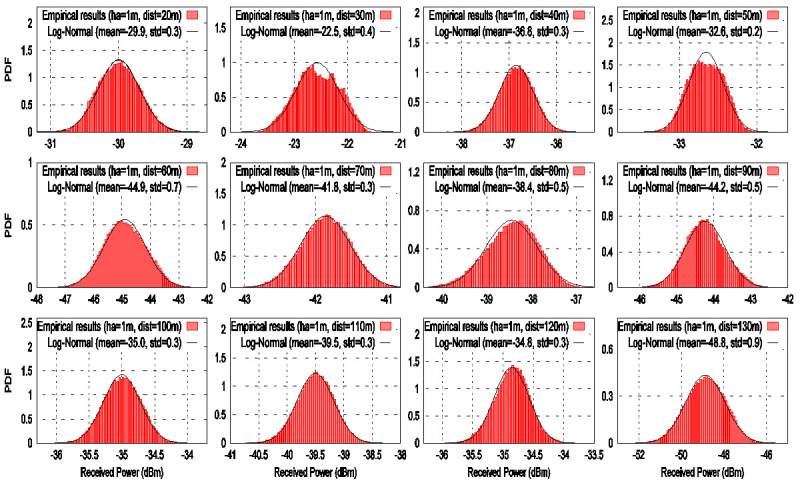
Histograms at each distance from 20 m to 130 m for a height of 1 m over the river within vegetation.

**Figure 11 sensors-21-06872-f011:**
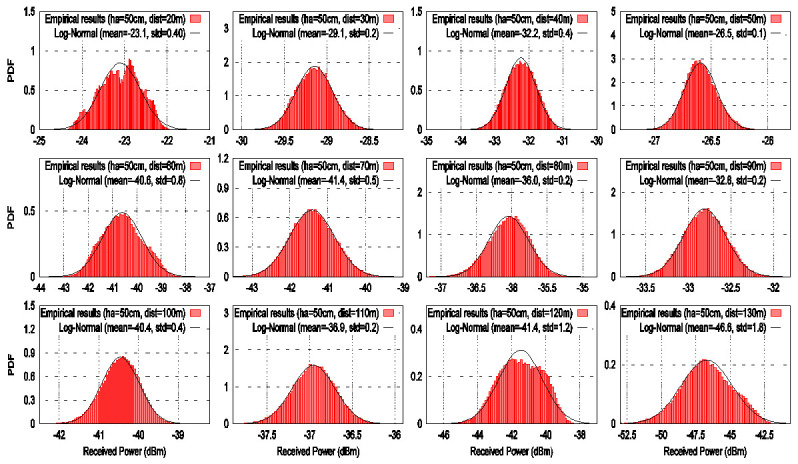
Histograms at each distance from 20 m to 130 m for a height of 50 cm over the river within vegetation.

**Figure 12 sensors-21-06872-f012:**
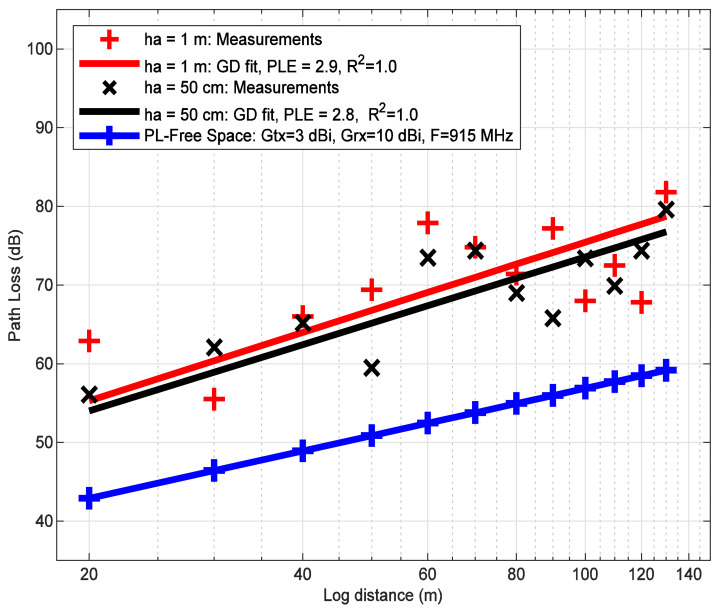
Obtaining the empirical path loss exponent (PLE) by linear regression. The gradient descent technique was used to get the PLE, in addition to computing the coefficient of determination at distances from 20 m to 130 m for both antenna heights (i.e., 1 m and 50 cm over the water).

**Figure 13 sensors-21-06872-f013:**
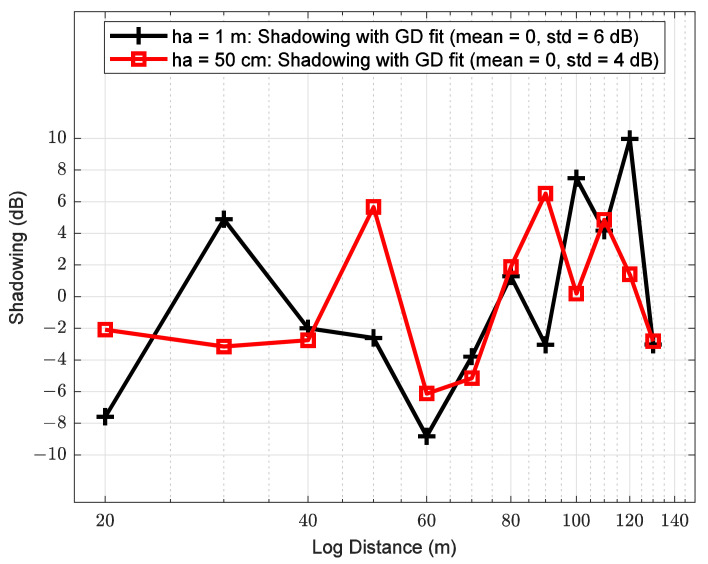
Shadowing between transmitter and receiver in a river within vegetation for both antenna heights of 1 m and 50 cm.

**Figure 14 sensors-21-06872-f014:**
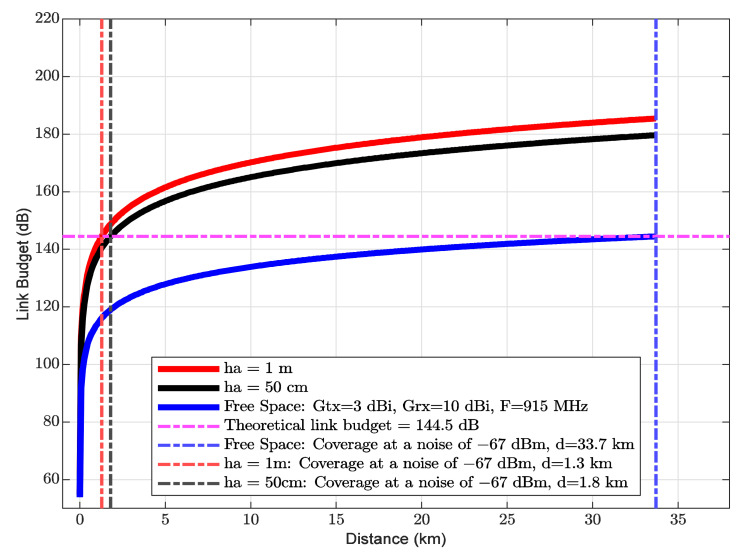
Link budget of the experimental path loss models at ha of 1 m and ha of 50 cm and the theoretical free space path loss model. In addition, the coverage reached by the path loss models analyzed over rivers within vegetation.

**Table 1 sensors-21-06872-t001:** Comparison of the propagation studies of the LoRa technology over water.

**Reference**	[17]	[22]			[23]	[24]		[25]
**Coverage (km)**	30	22		28	4	12.96		0.4
**Frequency (MHz)**	868 LOS	433 LOS	868 LOS	433 NLOS	433 NLOS	868 LOS		433 LOS
**PL-d0 (dB)**	128.95	X	X	X	X	X	X	X
**d0 (m)**	1000	X	X	X	X	X	X	X
**PLE**	1.76	X	X	X	X	X	X	X
**TAH (m)**	2	Not mentioned	Not mentioned	Not mentioned	3	2.1	3.5	1.5
**RAH (m)**	24	Not mentioned	Not mentioned	Not mentioned	0.8	13.2	13.2	4
**Scenario**	Sea	Sea	Sea	Sea	Sea	Sea	Sea	Sea
**RSSI Calibration**	Not used	X	X	X	X	X	X	X

**Table 2 sensors-21-06872-t002:** Comparison of propagation studies of the LoRa technology in vegetated environments.

**Reference**	[10]	[15]			[16]		[27]	[28]	[29]	[30]		[31]
**Coverage (km)**	0.71	47			1.5 (DR3)	1.6 (DR0)	0.25	X	1.75	0.23	1	0.05–0.09
**Frequency (MHz)**	868	868 NLOS	868 LOS		915 NLOS		915 NLOS	433 NLOS	868 NLOS	915 NLOS	868 NLOS	868 NLOS
**PL-d0 (dB)**	95.5	111	Not mentioned		61.1	55.3	X	X	X	X	X	X
**d0 (m)**	1	Not mentioned			Not mentioned		X	X	X	X	X	X
**PLE**	2.0	3.0	1.9		1.0	1.1	X	X	X	X	X	X
**TAH (m)**	1.5	0.2	3	1.5	<2	<2	X	0.5, 1.3, 2, 2.5, 3	2	1	1	X
**RAH (m)**	1.5	70	70		<2	<2	X	1.3	2.5	1	1	X
**Scenario**	Forest	Rural with trees			Rural with trees		Forest	Forest	Forest	Forest	Forest near lake	Vegetation
**RSSI Calibration**	Not used	Not used			Not used		X	X	X	X	X	X

**Table 4 sensors-21-06872-t004:** Empirical constants for MED models.

Model	A	B	C	Conditions
Weissberger [58]	1.33	0.284	0.588	$14<d\left[m\right]\leq 400$ $0.23\leq f\left[\mathrm{GHz}\right]\leq 95$
	0.45	0.284	1	$0\leq d\left[m\right]<14$ $0.23\leq f\left[\mathrm{GHz}\right]\leq 95$
ITU-R [59]	0.2	0.3	0.6	$0<d\left[m\right]<400$ $200\leq f\left[\mathrm{MHz}\right]\leq 95,000$
COST 235 [60]	26.6	−0.2	0.5	out-of-leaf $200\leq f\left[\mathrm{MHz}\right]\leq 95,000$
	15.6	−0.009	0.26	in-leaf $200\leq f\left[\mathrm{MHz}\right]\leq 95,000$
FITU-R [61]	0.37	0.18	0.59	out-of-leaf $200\leq f\left[\mathrm{MHz}\right]\leq 95,000$
	0.39	0.39	0.25	in-leaf $200\leq f\left[\mathrm{MHz}\right]\leq 95,000$

**Table 5 sensors-21-06872-t005:** Empirical parameters recollected from ITU P.833-9 [62].

Frequency (MHz)	A_1_ (dB)	α_1_	γ dBm	Condition
105.9–2117.5	1.37	0.42	0.2	Mixed forest (height 14 m) h_rx_ = 1.5 m

**Table 6 sensors-21-06872-t006:** SNR and background noise power (P_N_) for each ha of 1 m and 50 cm at each distance.

d (m)	20	30	40	50	60	70	80	90	100	110	120	130
*ha* = 1 m: *P_N_* (dBm)	−69.2	−67.2	−69.1	−67.2	−66.8	−69.1	−67.9	−68.9	−68.9	−67.9	−68.0	−68.2
*ha* = 1 m: SNR (dB)	39	45	32	35	22	27	30	25	42	28	33	19
*ha* = 50 cm: *P_N_* (dBm)	−66.5	−64.8	−65.9	−64.3	−67.7	−63.7	−66.9	−66.1	−64.3	−66.7	−60.8	−62.3
*ha* = 50 cm: SNR (dB)	43	36	34	38	27	22	31	33	24	30	19	16

**Table 7 sensors-21-06872-t007:** Parameters of the experimental path loss models.

Environment	Antenna Height (m)	PLE	PL(d_0_ = 1m) (dB)	*σ* (dB)	*R* ^2^	Iterations GD Technique	Learning Rate
River within vegetation	0.5	2.8	17.6	4	1	30,000	0.0001
River within vegetation	1	2.9	17.8	6	1	61,000	0.0001

**Table 8 sensors-21-06872-t008:** Parameters to evaluate the link budget of the experimental and theoretical path loss models.

Parameters	Value
Noise measured in the river within vegetation	−67 dBm
Distance	From 1 m to 33.7 km
Antenna transmitter gain	3 dBi
Antenna receiver gain	10 dBi
Transmitter power	20 dBm
Frequency	915 MHz
Noise figure	6 dB
Bandwidth	125 kHz

**Table 9 sensors-21-06872-t009:** Statistics of comparison between the theoretical FSPL model and the proposed models over water within vegetation.

Theoretical Model	MAPE (%)	
	Experimental Models	
	*ha* = 1 m, *σ* = 6 dB	*ha* = 50 cm, *σ* = 4 dB
FSPL	31	28
FSPL + ITU-R P.2108-0	15	11
FSPL + FITU-R (in-leaf)	12	17
FSPL + ITU-R P.833-09	14	10

**Table 10 sensors-21-06872-t010:** Comparison between the theoretical two-ray model and the proposed models over water within vegetation.

Theoretical Model	MAPE (%)	
	Experimental Models	
	*ha* = 1 m, *σ* = 6 dB	*ha* = 50 cm, *σ* = 4 dB
Two Rays	15	24
Two Rays + ITU-R P.2108-0	31	46
Two Rays + FITU-R (in-leaf)	58	74
Two Rays + ITU-R P.833-09	32	46

## Data Availability

Publicly available datasets were analyzed in this study. This data can be found here: [https://repositoriotelecomfi.unam.mx/index.php/s/pHmrpzJJ8qMb7px].

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
