# Peer review of "A Propagation Study of LoRa P2P Links for IoT Applications: The Case of Near-Surface Measurements over Semitropical Rivers"

_sensors, 2021, doi:10.3390/s21206872_

Round 1
Reviewer 1 Report
As the authors have stated at the conclusions: This paper proposes a new methodology to obtain propagation models by using the power of pure chirp signals from a LoRa ED radio.
The paper has a good background and practical results, but it lacks originality and pragmatism. The title and abstract do not describes well the paper. IoT (normally used in a massive data transmission scenario is not the case). The place in the title (semitropical rivers in Mexico) is also unnecessary and meaningless. The “novelty “should lay down on the model, not at one specific place.
It is necessary to describe the altitude difference between the route campaign figure 4. It also should be part of the model. Different heights should impact on the optimal setup.
Two antenna heights are insufficient or such study, since it is one important input in the model. Also, the model should consider that and infer what should be the optimum height for both T and R.
The optimization approach LR using GD is not the best in this situation. The problem is nonlinear multimodal, and the results could be stuck in local optima.
The distance ha should also consider the altitude difference between the points.
The mathematical presentation needs more formalism.
Author Response
Reply to the Review Report

Reviewer 2 Report
Generally speaking, it's quite pleasant to read through this paper that has obvious novelties with implementation realized. The overall quality of this paper is very good, but there are some questions and concerns as follows for authors to consider:
1. Some more background literature might be considered at the beginning, such as,
[1] R. Sinha, Y. Wei, S. Hwang, “A survey on LPWA technology: LoRa
and NB-IoT”, ICT Express 3, Korean Institute of Communications
Information Sciences, March 2017
[2] G. H. Baracat and J. M. C. Brito, "NB-IoT Random Access Procedure Analysis," 2018 IEEE 10th Latin-American Conference on Communications (LATINCOM), 2018, pp. 1-6, doi: 10.1109/LATINCOM.2018.8613207.
[3] Y. Huo, X. Dong and S. Beatty, "Cellular Communications in Ocean Waves for Maritime Internet of Things," in IEEE Internet of Things Journal, vol. 7, no. 10, pp. 9965-9979, Oct. 2020, doi: 10.1109/JIOT.2020.2988634.
[4] D. Madeo, A. Pozzebon, C. Mocenni and D. Bertoni, "A Low-Cost Unmanned Surface Vehicle for Pervasive Water Quality Monitoring," in IEEE Transactions on Instrumentation and Measurement, vol. 69, no. 4, pp. 1433-1444, April 2020, doi: 10.1109/TIM.2019.2963515.
2. Water waves in the rivers, and also rain might cause some problems for the radio signal propagation, for your case, have you seen some problems.
Author Response
Reply to the Review Report

Reviewer 3 Report
I think this paper is excellent and provides a really great measurement modeling analysis. However I have the following concerns:
- Figure 7 is redundant, either zoom in showing an actual transmission versus time or remove thsi figure as it provides no importance
- I am not convinced with the methodology that the author utilizes to calculate the received power. The methodology is correct assuming perfect conditions (i.e. no noise and no interference). However, in a practicalo setting detection theory must be implemented, i.e. calculate the probability of false alarm and from that detection threshold and so on. As it stands, this assumption is not correct for a practical setting such as the one this methodology was utilized for.
- Figure 8: please use time as the x-axis and not the number of samples
- I would be interested if these signals can be compared to simulation scnearios such as the ones performed in this paper: https://ieeexplore.ieee.org/stamp/stamp.jsp?arnumber=9395074
- Needs to have a comparison with state-of-the-art propagation models by ITU or major papers such as:
- Kyosti, Pekka, Juha Meinila, et al. WINNER II Channel Models. D1.1.2 V1.2. IST-4-027756 WINNER II, September 2007.
- https://ieeexplore-ieee-org.ezproxy.lib.rmit.edu.au/document/7914696
- https://ieeexplore-ieee-org.ezproxy.lib.rmit.edu.au/document/670736
Author Response
Reply to the Review Report

Round 2
Reviewer 3 Report
Paper is in good shape now, Thank you